# Diagnostic and Interventional Radiology Management of Ureteral Iatrogenic Leakage after Gynecologic Surgery

**DOI:** 10.3390/diagnostics11050750

**Published:** 2021-04-22

**Authors:** Federico Fontana, Filippo Piacentino, Christian Ossola, Jvan Casarin, Andrea Coppola, Antonella Cromi, Anna Maria Ierardi, Gianpaolo Carrafiello, Antonio Basile, Federico Deho, Fabio Ghezzi, Giulio Carcano, Massimo Venturini

**Affiliations:** 1Diagnostic and Interventional Radiology Department, Ospedale Di Circolo, ASST dei Sette Laghi, 21100 Varese, Italy; filippo.piacentino@asst-settelaghi.it (F.P.); andrea.coppola@asst-settelaghi.it (A.C.); massimo.venturini@uninsubria.it (M.V.); 2School of Medicine and Surgery, Università degli Studi dell’Insubria, 21100 Varese, Italy; c.ossola7@gmail.com (C.O.); jvan.casarin@asst-settelaghi.it (J.C.); Antonella.Cromi@asst-settelaghi.it (A.C.); Fabio.Ghezzi@asst-settelaghi.it (F.G.); giulio.carcano@uninsubria.it (G.C.); 3Obstetrics and Gynecology Department, Ospedale “Filippo Del Ponte”, ASST dei Sette Laghi, 21100 Varese, Italy; 4Diagnostic and Interventional Radiology Department, Fondazione IRCCS Cà Granda Ospedale Maggiore Policlinico, 20122 Milan, Italy; amierardi@yahoo.it (A.M.I.); gcarraf@gmail.com (G.C.); 5Department of Radiology and Department of Health Sciences, Fondazione IRCCS Cà Granda Ospedale Maggiore Policlinico and University of Milano, 20122 Milan, Italy; 6Radiology Unit I, Department of Medical Surgical Sciences and Advanced Technologies, University Hospital “Policlinico-Vittorio Emanuele”, 95123 Catania, Italy; basile.antonello73@gmail.it; 7Urology Department, Ospedale Di Circolo, ASST dei Sette Laghi, 21100 Varese, Italy; federico.deho@asst-settelaghi.it; 8Surgery Department, Ospedale di Circolo, ASST dei Sette Laghi, 21100 Varese, Italy

**Keywords:** urinary leak diagnosis, interventional uroradiology, ureteral leakage, iatrogenic ureteral injuries

## Abstract

Objective: To report safety and efficacy of interventional radiology procedures in the treatment of gynecologic iatrogenic urinary leaks. Methods: A retrospective analysis of iatrogenic ureteral lesions treated between November 2009 to April 2019 was performed. Under ultrasound (US) and fluoroscopy guidance, an attempt to place a ureteral stent and nephrostomy was carried out in the same session using an anterograde percutaneous approach. At the end of any procedure, a fluoroscopic control and a cone-beam CT scan (CBCT) were performed to check the correct placement and functioning of the nephrostomy and DJ stent. In cases of difficult ureteral stent placement via the single anterograde approach, the collaboration of urologists was requested to perform a rendezvous technique, combined with the retrograde approach. Results: DJ stent placement was achieved using the anterograde approach in 12/15 (80.0%) patients and using the retrograde approach in 3/15 cases (20.0%). Moreover, in 3/15 (20.0%) patients, surgical treatment was needed: in one case because of the persistence of ureteral stenosis at 6 months, and in the other two cases due to ureter-vaginal fistula. No major complications were recorded; overall, minor complications occurred in 4/8 patients. Conclusion: Percutaneous minimally invasive treatment of iatrogenic ureteral lesions after gynecological surgery is a safe and effective option.

## 1. Introduction

Although extremely rare, iatrogenic ureteral injuries are a severe complication of gynecological surgery [1]. The risk of damage increases when the normal anatomy is altered by primary pathologic factors, by pelvic adhesions, or when the ureter is poorly recognizable because of intraoperative complications, such as severe bleeding [2]. Urinary tract injuries occur in 0.2%–1% of all gynecologic pelvic surgeries, with higher risk reported in case of severe endometriosis and locally advanced cervical cancers [3]. Moreover, uterine arteries cross the ureters anteriorly, with higher risk of iatrogenic injury during hysterectomy, ranging from 1% in in laparoscopic approach for benign disease, to 10.7% in open surgery for cervical malignancy. Injuries occur most frequently in the lower third of the ureter (51%), and less frequently in the upper and middle third (30% and 19%, respectively) [4]. Injuries can occur by ligation or kinking by a ligature, by clamping, division, devascularization, or diathermy-related injury; however, the most common injury mechanism is complete or partial transection [5,6].

Although intra-operative identification and repair of ureteral injuries is associated with better outcomes, 50%–70% of these lesions are missed in acute settings [7]. Patients with unrecognized injuries can show abdominal pain, fever, anuria, peritonitis, and even vaginal urinary leaking [8]. In these patients, radiologists play a pivotal role in both the diagnosis of the injury and tailoring of treatment [9].

Computed tomography urography (CTU) is the study of choice in the diagnosis of urine leaks and urinomas, demonstrating the entity of the leak and ureteral stumps [10]. Signs of ureteral injury include extraluminal contrast medium, hydronephrosis, ureteric obstruction, urinary ascites, and localized fluid collections such as urinoma [11].

Despite the clinical importance of this condition, no shared and codified guidelines about the treatment of this type of injury were able to be found in medical literature.

The purpose of this study is to report the safety and efficacy of interventional radiology procedures in the treatment of iatrogenic urinary leaks in different types of ureteral lesions in gynecologic surgery.

## 2. Materials and Methods

### 2.1. Patients

A retrospective analysis of prospectively collected procedures achieved at our Interventional Radiology Unit on patients sent from the Gynecology and Obstetrics Department with iatrogenic ureteral lesions between November 2009 to April 2019 was performed. Demographic characteristics and clinical findings were retrospectively obtained from patients’ medical records and institutional procedure forms.

### 2.2. Diagnosis

CTU (Aquilion 64, Toshiba Medical Systems, Ōtawara, Tochigi, Japan) was obtained for all patients to diagnose the ureteral lesion and for procedure planning. Diagnosis timing, early or delayed, was calculated. Ureteral leakage site (proximal, middle, distal), side (right or left), and hydronephrosis presence (mild, moderate) or absence were also evaluated.

### 2.3. Technique

Coagulation parameters were corrected for INR <1.5 and platelets >50,000/mm^3^.

Written informed consent was obtained from each patient before the procedure.

Our method was based on the initial nephrostomy and the attempt to implant a DJ stent in the same session.

All procedures were performed in the angiography suite under ultrasound (US) and fluoroscopic guidance (iU22, Philips Healthcare, Best, The Netherlands; AlluraXper FD20, Philips Healthcare, Best, The Netherlands), under local anesthesia (Mepivacain 2%, 10 mL) and moderate sedation (Propofol/Fentanyl/Ketamine in appropriate doses, according to operator preference).

#### 2.3.1. Anterograde Approach

All procedures were performed with the patient in the prone position. Using US guidance, a Chiba needle was used to access the renal cavity, preferably from the lower or middle posterior calyx. An 18 G needle (Chibell, Byopsybell, Mirandola, Modena, Italy) was used in markedly dilated, and a 21 G needle (Accustick, Boston Scientific, Marlborough, MA, USA) in mildly or non-dilated collecting systems. After the placement of a short introducer (5-F, Cordis, Miami, FL, USA), a descending pyelography study was performed to confirm the presence and the site of the leakage. Subsequently multiple attempts were usually performed to cross the damaged tract with a 0.035-inch hydrophilic guide-wire (Glidewire, Terumo, Tokyo, Japan). In case of successful crossing, over a stiff guide (Amplatz, Cordis, Miami, FL, USA) a plastic 8-F double-J stent (Flexima Ureteral Stent, Boston Scientific, Marlborough, MA, USA) was placed (Figure 1A–D). An 8F nephrostomy tube (Boston Scientific, Marlborough, Massachusetts, USA) was maintained in all cases.

#### 2.3.2. The Rendezvous Technique

In the cases of the anterograde approach that failed to pass the ureteral damaged tract, only the nephrostomy tube was placed and a second attempt with the collaboration of urologists was performed combining anterograde and retrograde approaches (rendezvous technique), with the patient in the supine position. A retrieval 5-F goose-neck snare catheter with a 30 mm loop (Hooker, Meditalia, Biomedica, Modena, Italy) was advanced from the nephrostomy access in order to reach the ureter at the level of the iatrogenic injury. Then, the urologist from the bladder cannulated the ureteric orifice using a flexible ureteroscope, running through the distal ureteral stump and trying to bring a hydrophilic guidewire as close as possible to the system previously positioned from the other side. The guide from the bladder approach was then caught and pulled out from the goose neck catheter through the nephrostomy access. After the change of the hydrophilic guide with a stiff guidewire and applying bidirectional traction on the guide, a double-J stent was finally placed.

Finally, the nephrostomy therefore was left in situ for gravity drainage.

#### 2.3.3. Ureteral Metallic Stenting

In one case of a ureterovaginal fistula, after keeping the DJ stent-nephrostomy system for a 3 month period, given the fistula persistence, an 8 mm × 80 mm metallic coated removable stent (Allium Medical Solutions, Caesarea, Israel; Figure 2A,B) was placed. Three months after the stent placement, because of fistula persistence, the patient underwent surgery.

### 2.4. Patient Discharge and Follow-Up

Clinical and radiological resolution after interventional radiological management was investigated. At the end of the procedure and 7 days later, a pyelographic fluoroscopic control and a cone-beam CT scan (CBCT) were performed to check the correct placement and functioning of the nephrostomy and DJ stent, and to exclude iodine spillage. The nephrostomy tube was then removed.

Stent replacement/removal was managed in the angiography suite using a retrograde technique (Figure 3A–D) as previously described [12] at 1, 3 and 6 months, and then every 6 months. A retrograde pyelography and a CBCT scan were performed in all cases to verify the restored integrity of the injured ureter at every control after DJ stent removal before making a decision about for a new DJ stent placement; ureteral continuity was ensured by a guidewire during this diagnostic phase.

Creatinine, C reactive protein (CRP), and white blood cell count before and after every interventional procedure were assessed.

All complications were recorded and classified as minor and major according to Common Terminology Criteria for Adverse Events (CTCAE), Version 4.0 [13].

## 3. Results

Fifteen female patients (mean age 46.1 years, range 32–67 years) were enrolled in this retrospective study. Eleven were surgically treated for benign disease (endometriosis in six cases, fibromatosis in five cases), four for malignant disease (advanced cervical cancer in all four cases). The laparoscopic approach was used in 11/15 cases, and the remaining four cases underwent laparotomic surgery.

Diagnosis was intra-operative in three cases, in the recent post-operative time (from 4 to 26 days after surgery) in 11 cases, and in a delayed postoperative time (75 days after surgery) in one case. Regarding the site of the ureteral leakage, the distal segment was involved in 12 of 15 patients (80.0%) and the middle in 3/15 (20.0%). The involved side was right in 7/15 (46.6%) patients and left in 8/15 (53.4%). No hydronephrosis was recorded in 8/15 (53.4%) patients, whereas it was found in 7/15 (46.6%) patients (mild in four patients, moderate in three).

The anterograde approach was successful in treating 12/15 (80.0%) patients.

In 3/15 (20.0%) patients, because of the misalignment of the ureteral stumps, the rendezvous technique was needed, with successful positioning of the double-J stent.

In 3/15 (20.0%) patients, surgical treatment was needed: in one case because of the persistence of ureteral stenosis at 6 months, and in the other two cases due to ureter-vaginal fistula (Figure 2A,B).

In 8/15 (53.4%) cases, the stent was completely removed without leakage persistence and the mean duration of catheterization was 6 months (range 3–12 months); specifically, in 6/8 (75.0%) cases without stent replacement (mean catheterization of 4.5 months), whereas in the remaining 2/8 (25.0%) cases, one stent replacement was performed (mean catheterization time of 10.5 months).

At the end of the follow up (FU) period, the stent was still in place in 4/15 (27%) cases; specifically, in 2/4 due to disease recurrence (1/2 with advanced cervical cancer and 1/2 for deep pelvic endometriosis), and in 1/4 to prevent complication because of simultaneous radiotherapy treatment in advanced cervical cancer. In addition, the stent was still in place in 1/4 patients with the persistence of minimum ureteral stenosis due to refusal of surgery.

An algorithm to summarize our management of iatrogenic ureteral injuries is shown in Figure 4.

No major complications were recorded; minor complications occurred in 4/15 (26.7%) cases during the follow up. In particular, two patients had an infectious complication, resolved in both cases with an adequate antibiotic therapy and stent replacement. In the other two patients who were candidates for stent removal, to relieve a ureteral stenosis in correspondence with the damage site, it was recommended to perform a balloon dilatation and prolong the maintenance of the stent.

All case data are summarized in Table 1.

## 4. Discussion

The majority of iatrogenic ureteral injuries occur during gynecological procedures [14] and it is estimated that 52%–82% of iatrogenic injuries occur during gynecologic surgery [15,16].

Ureteral injury is diagnosed intraoperatively only in 8.6% of cases [17]. Ureteral trauma must be suspected in the postoperative period when facing upper urinary tract obstruction, urinary fistulae, or sepsis [18].

Postoperative diagnosis of ureteral lesions can be very challenging because clinical manifestations are often non-specific [1].

When ureteral trauma is suspected, CT urography (CTU) is considered the gold standard tool for diagnosis according to the European Association of Urology (EAU) [19] and the American Urological Association (AUA) guidelines [20]. In fact, CTU allows the demonstration of ureteral contrast extravasation, urinoma, ascites, and fistulae, and can be used to study the entity of the lesion to guide the treatment planning [21,22]. Retrograde or anterograde pyelography are also sensitive radiographic tests for ureteral injury, while also allowing the simultaneous stenting procedure [10,14].

Despite the clinical importance of ureteral lesion management, the quantity of scientific work reported about this topic is relatively low in the medical literature.

Controversy remains about the ureteral repair strategy regarding surgical or conservative approaches, particularly urological endoscopic or interventional radiology procedures, or both in association (the “rendezvous” technique) [16,23].

The first success of a percutaneous procedure to treat iatrogenic leakage was reported by Druy et al., in 1984, successfully treating five patients with post-traumatic ureteral anastomosis dehiscence [23]. In 1995, Lask and Colleagues [24] indicated that for patients treated with percutaneous nephrostomy, complete spontaneous recovery of the injured ureter occurred in 80% of cases.

In our experience, in case of ureteral lesion, an anterograde approach was attempted to position in the same session both ureteral stent and nephrostomy; in case of failure, we opted for the rendezvous technique combining anterograde and retrograde approaches. Macrì et al. [25] demonstrated that the rendezvous technique increases the success rate of anterograde ureteral stenting in difficult cases from 78.6% to 88.1% making it a valid option in case of failure of conventional ureteral stenting.

Recently, Zini et al. [26] also described the reliability of this approach in complete iatrogenic ureteral section, especially in oncological patients. Liu et al. [27] reported a series of eight patients with iatrogenic total section of the ureter treated by endoscopic rendezvous using a flexible ureteroscope cranially and a rigid ureteroscope caudally, achieving the repair of the urinary tract in all patients [27].

In our study, the aim of restoring the integrity of the ureter was achieved in 12/15 patients (80%), consistent with data reported in the medical literature (64.7%–100%) [1,16,28]. Balloon dilatation was required in 4/15 patients due to iatrogenic strictures, always successfully.

In 2003, Reddy et al. described the use of covered stents in the treatment of a uretero-cutaneous fistula in a patient with ureteral injury following surgical repair of a left iliac artery aneurysm [29]. We reported a case of ureterovaginal fistula, treated with a metallic coated removable stent given the failure of healing after 6 months using a DJ stenting–nephrostomy system.

In our study, the minimally invasive strategy was proven to be effective in the treatment of partial ureteral lesion associated with urinoma. Moreover, in patients with early leakage resolution, no stent replacement was needed and no procedural complications were reported. However, the persistent leakage of a fistulous output makes conservative treatment less valid [21,22].

In our experience, stent replacement was managed by a trans-urethral technique in an angiosuite, with light sedation, using a gooseneck system and 8 Fr double J stent [12].

No specific recommendation exists on the optimal stenting duration and FU [14,30,31]; in our study, the mean stenting duration was 6 months. Long-term follow-up with CTU, renal US, and serum creatinine can be performed at 3–6 months and repeated at 12 months in order to detect late strictures and complication [32].

We performed 1 week and 1 month follow up after ureteral stenting by anterograde pyelography and CBCT to verify the restored integrity of the injured ureter and the possible appearance of complications; subsequent checks were performed 3 months following the procedure in all patients and every 3 months up to 12 months. The stent was still in place in four of 15 patients at the end of the follow up period, without evidence of leakage.

Our study has some limitations because its retrospective nature limited the cohort of patients; moreover, the management of iatrogenic ureteral injuries is controversial due to the lack of guidelines.

## 5. Conclusions

Iatrogenic ureteral lesions are a non-negligible surgical complication, often detected postoperatively or after a delay, resulting in substantial morbidity.

Radiologists play a key role in both the diagnosis of the injury and guiding or carrying out the treatment.

The conservative interventional radiological approach can represent the first choice for traumatic ureteral section treatment, especially in hospitals without urologist assistance and as an approach to the injuries identified in the acute phase. In the complete ureteral section, after anterograde technique failure, the second step can be represented by a combined radiological–endoscopic (“rendezvous”) technique, limiting the use of surgery to selected cases.

In our limited experience, a minimally invasive treatment of iatrogenic ureteral lesions after gynecological surgery was a safe, manageable, and effective option. However, further studies including a larger cohort of patients will be necessary to support these preliminary conclusions.

## Figures and Tables

**Figure 1 diagnostics-11-00750-f001:**
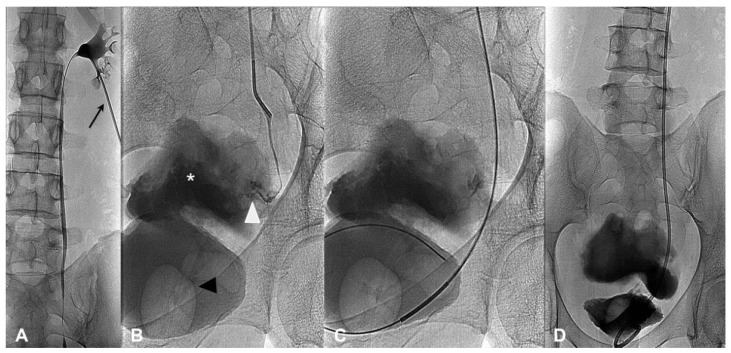
(**A**–**D**)—A successful anterograde approach. Right descending pyelography (**A**,**B**) performed using a 5 Fr introducer (black arrow) proves a lesion (white arrow head) of the distal third of the right ureter associated with urinoma (white asterisk); the bladder is distended by contrast medium injected through a urinary catheter (black arrow head). (**C**) A hydrophilic guidewire is advanced in the bladder with a catheter through the ureteral distal stump and a double-J stent (**D**) was implanted and correctly positioned.

**Figure 2 diagnostics-11-00750-f002:**
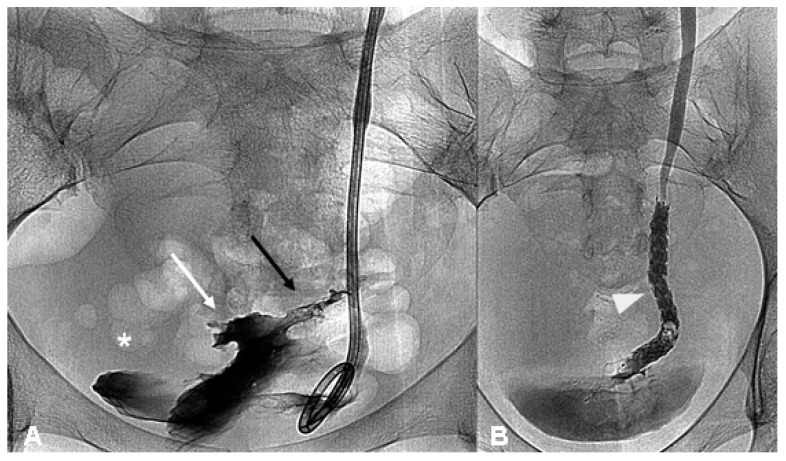
(**A**,**B**)—A persistent fistula treated with ureteral stent. Left descending pyelographic study demonstrates a ureterovaginal fistula (black arrow) in the presence of a right DJ stent with opacification of both vaginal lumen (white arrow) and the bladder (asterisk) (**A**); removable metallic coated stent (white arrow head) is positioned without evidence of a fistula (**B**).

**Figure 3 diagnostics-11-00750-f003:**
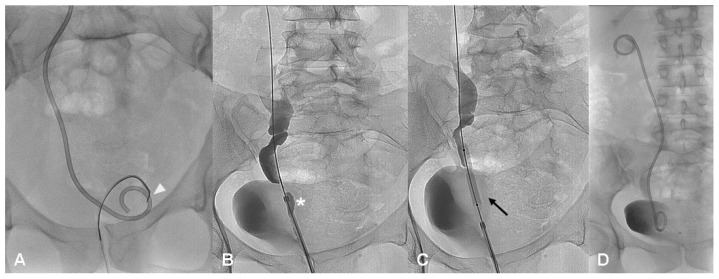
(**A**–**D**)—Double J stent substitution and ureteral ballooning. The distal end of the right ureteral double J stent is grasped with a goose-neck snare catheter (white arrow head) positioned in the lumen of the opacified bladder under fluoroscopic guidance (**A**); ascending pyelography through a vascular 7 Fr introducer shows a residual stenosis (white asterisk) at the distal portion of the right ureter (**B**); ureteral dilatation procedure is performed using 6 × 40 mm balloon (black harrow) (**C**); a new double-J stent was inserted and correctly positioned (**D**).

**Figure 4 diagnostics-11-00750-f004:**
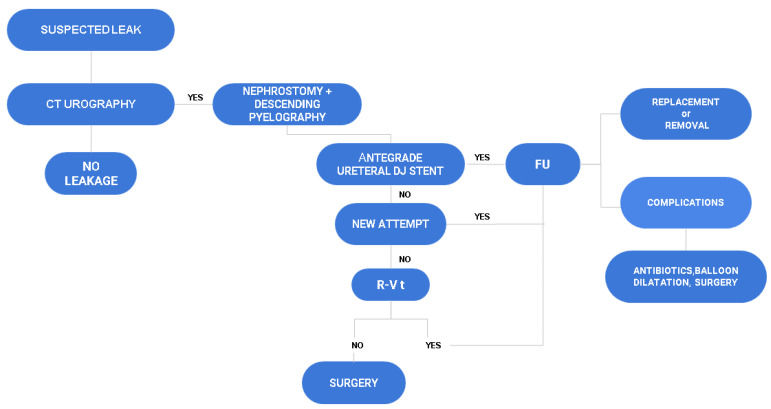
Management algorithm for iatrogenic ureteral injuries. CT: computed tomography; DJ: double J; FU: follow up; R-V t: rendezvous technique.

**Table 1 diagnostics-11-00750-t001:** Case data.

Case	Age (Years)	Gynecological Pathology	Gynecological Surgery	Side	Ureteral Injury Site	US Kidney Dilatation	Diagnosis	Injury Type	Ureteral Stent Implantation	Stent Specifications (F × cm)	Time of Leakage Resolution (Weeks)	N of Stent Replaced	Complications	Stent FU
1	35	UFM	Laparoscopy	Right	Distal	no	Intra-operative	Partial	Retrograde	8 × 24	1 week	None	None	Removed after 3 months
2	35	DPE	Laparoscopy	Left	Medial	no	Post-operative	Complete	RVt	8 × 26	1 week	None	Acute pyelonephritis	Removed after 6 months
3	59	UFM	Laparotomy	Right	Medial	moderate	Intra-operative	Partial	Retrograde	8 × 26	6 months	3	Acute pyelonephritis + recurrent stricture	Removed after 12 months + surgery
4	63	UFM	Laparoscopy	Left	Distal	mild	Post-operative	Partial	Anterograde	8 × 24	3 months	2	None	In progress (refusal)
5	32	DPE	Laparoscopy	Left	Distal	no	Intra-operative	Partial	Anterograde	8 × 24	3 months	1	Stricture	Removed after 9 months
6	41	ACC	Laparoscopy	Right	Distal	no	Post-operative	Complete	RVt	8 × 26	No resolution after 6 months	None	None	Removed after 6 months + surgery (UVF)
7	47	DPE	Laparoscopy	Right	Distal	no	Post-operative	Complete	RVt	8 × 26	1 month	None	None	Removed after 3 months
8	33	DPE	Laparoscopy	Left	Distal	no	Post-operative	Partial	Anterograde	8 × 26	1 month	None	None	Removed after 3 months
9	42	UFM	Laparoscopy	Left	Distal	no	Post-operative	Partial	Anterograde	8 × 26	3 months	None	None	Removed after 6 months
10	48	ACC	Laparotomy	Left	Medial	mild	Delayed	Partial	Anterograde	8 × 24	6 months	3	None	In progress (Rtp)
11	67	ACC	Laparotomy	Right	Distal	moderate	Post-operative	Partial	Retrograde	8 × 24	6 months	1	Stricture	Removed after 12 months
12	41	DPE	Laparoscopy	Right	Distal	no	Post-operative	Partial	Anterograde	8 × 26	3 months	4	None	In progress (relapse)
13	57	ACC	Laparotomy	Right	Distal	mild	Post-operative	Partial	Anterograde	8 × 24	6 months	2	None	In progress (relapse)
14	52	UFM	Laparoscopy	Left	Distal	mild	Post-operative	Partial	Retrograde	8 × 24	3 months	None	None	Removed after 6 months
15	40	DPE	Laparoscopy	Left	Distal	moderate	Post-operative	Partial	Anterograde	8 × 24 + MCS	12 months	2	None	Removed after 12 months + surgery (UVF)

ACC: advanced cervical cancer; DPE: deep pelvic endometriosis; FU: follow up; MCS: metallic coated stent; RTp: radiotherapy; RVt: rendezvous technique; UFM: uterine fibromatosis; US: ultrasound; UVF: ureterovaginal fistula.

## Data Availability

The data presented in this study are available in Table 1.

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
