# Peer review of "Diagnostic and Interventional Radiology Management of Ureteral Iatrogenic Leakage after Gynecologic Surgery"

_diagnostics, 2021, doi:10.3390/diagnostics11050750_

Round 1

Reviewer 1 Report

The authors report an interesting case series focusing on interventional radiology management of ureteral complication of gynecological surgery. Despite the methods applied should represent the standard of care in a tertiary care Hospital with limited novelty, in my opinion the message of the manuscript is of interest to the scientific community, especially to surgeons and physicians not familiar with interventional radiology treatments. 

General comments

Minor English language revision should be performed.  Some grammar and spelling errors throughout the text.

Specific comments

Abstract - Intro: Given the small number of patients and the retrospective nature of the study I would change "to evaluate" with "to report". Reliability is redundant in my opinion. Methods: I would detail inclusion criteria and general methodology rather than technical procedural details, that should be synthesized. Results: I would report technical and clinical success rates rather than "restoration of integrity" which is an ambiguous term. I understend technical success in 12/15. What about the other 3 cases?

Text

Introduction: OK. Sentences in lines 48-50 lack references. Please add. As suggested for the abstract I would change "evaluate" with "report". Reliability is redundant in my opinion.

Methods: the sentence "were prospectively collected for this retrospective study" is confusing and should be changed for example with "a retrospective analysis of prospectively collected radiological procedures...". In line 81 I would remove symptoms which are subjective. Lines 81-82 should be moved to line 163 with more accurate information about outcome evaluation.

Line 149: was CBCT performed in all cases? why was fluoroscopic control not considered enough?

Results: Clear and simple, OK. Lines 201-201: I'm not sure these can be considered complications but rather primary clinical failure with secondary clinical success.

Discussion and conclusion: OK

Reviewer 2 Report

Work well written and relatively well illustrated.
Very relevant topic. Treatment of relatively frequent surgical complications, with a very strong impact on the lives of patients in the short, medium and eventually long term.
It seems to me an option in hospitals without urologist assistance and as an approach to the injuries identified in the acute phase. This should be added to the conclusions.

Problems:
1. Why not always use cystoscopic control and try the retrograde catheter placement as the first approach?
2. Isn't percutaneous use more invasive than endoscopic use?
3. How did you control the patency of the ureteral lumen?
4. Follow-up after the procedure is not clear ...
5. Why prolonged catheterizations? (6 months on average?). What is the justification for so long ureteral catheterization?
6. What is the innovation compared to the current models of intervention in these cases?

Round 2

Reviewer 2 Report

None.